# A Novel Interprofessional Education and Supervision Student Placement Model: Student and Clinical Educator Perspectives and Experiences

**DOI:** 10.3390/ijerph191710734

**Published:** 2022-08-29

**Authors:** Priya Martin, Anne Hill, Martelle Ford, Tessa Barnett, Nicky Graham, Geoff Argus

**Affiliations:** 1Rural Clinical School, Faculty of Medicine, The University of Queensland, Toowoomba, QLD 4350, Australia; 2School of Health and Rehabilitation Sciences, The University of Queensland, Brisbane, QLD 4067, Australia; 3Cunningham Centre, Darling Downs Health, Toowoomba, QLD 4350, Australia; 4Cairns Hospital and Health Service, Cairns, QLD 4870, Australia; 5Southern Queensland Rural Health, The University of Queensland, Toowoomba, QLD 4350, Australia; 6School of Psychology and Counselling, University of Southern Queensland, Toowoomba, QLD 4350, Australia

**Keywords:** interprofessional education, interprofessional supervision, student placements, rural health, inter-sectoral partnerships

## Abstract

Interprofessional student placements can not only cater to the added pressures on student placement numbers but can also enhance the work readiness of new graduates. For rural areas, there is a potential for interprofessional student placements to attract the future healthcare workforce. However, tried and tested models of interprofessional placements in rural areas backed up by rigorous evaluation, remain scarce. The Rural Interprofessional Education and Supervision (RIPES) model was developed, implemented, and evaluated across four rural health services in Queensland to address this gap. Students from two or more professions undertook concurrent placements at RIPES sites, with a placement overlap period of at least five weeks. Eleven focus groups (*n* = 58) with clinical educators (CEs) and students were conducted to explore student and clinical educator experiences and perspectives. Content analysis of focus group data resulted in the development of the following categories: value of the RIPES placement model, unintended benefits to CEs, work units and rural areas, tension between uni-professional and IPE components, and sustainability considerations. Students and CEs alike valued the learning which arose from participation in the model and the positive flow-on effects to both patient care and work units. This unique study was undertaken in response to previous calls to address a gap in interprofessional education models in rural areas. It involved students from multiple professions and universities, explored perspectives and experiences from multiple stakeholders, and followed international best practice interprofessional education research recommendations. Findings can inform the future use and sustainability of the RIPES model.

## 1. Introduction

Facilitating interprofessional education with pre-entry healthcare students in clinical settings plays an important role in ensuring that students are practice-ready [1]. Interprofessional education (IPE) occurs when students from two or more professions learn together and from each other, to promote effective collaboration with the overall aim to improve health outcomes [2]. Rural clinical settings, where interprofessional education and collaborative practice (IPECP) opportunities remain limited [3], are optimal environments for students to gain exposure to a broad range of healthcare professions [3]. Embedding IPECP into rural healthcare settings allows students to receive IPECP experiences while also meeting increasing student placement demands [4,5]; and further encourage the recruitment of new graduates into rural healthcare services [6,7].

To date, there are limited high-quality studies on IPECP models in rural healthcare settings [7,8,9]. A study by McNair et al. (2005) implemented a fit-for-purpose IPE student placement model (RIPE) involving medical and nursing students in Victorian rural healthcare settings. This model involved an optional two-week placement, as an addition to the usual university course requirements. The findings from this study illustrated the benefits of this type of IPECP model in rural settings, such as improved interprofessional and teamwork skills in students [7]. However, there is a need to expand IPE models to other health professions in addition to nursing and medicine, plus integrate IPE into existing mandatory placement structures. IPE research too remains an area of attention with proposed international recommendations calling for more robust studies including the use of qualitative research methods, comparative studies, and participatory action research (PAR) [10,11]. PAR is a collaborative research approach between participants and researchers [12,13] that aims to generate thorough knowledge and implement change [14,15].

The Rural Interprofessional Education and Supervision (RIPES) model was developed to enhance pre-qualification student clinical education capacity in rural and remote services in the state of Queensland, Australia [16,17,18,19]. The RIPES model was also developed to strengthen much-needed academic and healthcare sector partnerships [20]. The model involved multiple students from different professions undertaking their uni-professional clinical placement at the same healthcare setting concurrently, overlapping by at least five weeks. RIPES was overlaid on the uni-professional placement in this period to enable participating students to undertake various structured IPE activities that were additional to their uni-professional placement requirements. Local clinical educators (CEs; also referred to as student supervisors) at the implementation site facilitated all the RIPES activities (including facilitating weekly tutorials, work shadowing and joint client sessions). The interprofessional supervision component involved CEs from other professions supervising a student while they undertook a joint client session or a joint home visit, with feedback mechanisms in place to communicate back to the profession-specific CE. Sites were prepared for IPECP which included training participating CEs in IPECP core concepts and competencies as per the Canadian Interprofessional Health Collaborative Framework [21] and IPE facilitation. All CEs were supported in developing and delivering IPE and interprofessional supervision activities contextualised to their site. Further information about the RIPES model, its development, a prior pilot phase, and a process evaluation have been reported elsewhere [16,17,18,19].

This study explored the enablers of and barriers to the implementation of the RIPES model from student and CE perspectives. Furthermore, it obtained in-depth information on the benefits and challenges of the model, to enable the future refinement and use of the model.

## 2. Materials and Methods

### 2.1. Research Design

A multi-site, multi-methods, pre-post, comparative design was used in the overall study. The qualitative component, reported here, utilised a participatory action research design. Focus groups were conducted separately with students and CEs at the completion of the RIPES placement. The quantitative component will be reported elsewhere.

### 2.2. Setting

This study was implemented in five work units across four health services in Queensland. Participating services were in Modified Monash Model 4 to 6 locations (MMM) [22]. The Modified Monash Model is a geographical classification system that targets health workforce programs by categorising metropolitan, regional, rural, and remote areas according to both geographic remoteness and town size [22]. The study had two components: this paper focuses on the qualitative data and findings. Findings from the quantitative component of this study will be reported separately.

### 2.3. Participants

Participants were post-qualification healthcare workers with clinical education/student supervision responsibilities, and health students from four professions: dietetics, occupational therapy, physiotherapy, and speech pathology. Participating sites were recruited via a state-wide organisational expression of interest process and consisted of teams delivering healthcare in regional and rural areas. Although recruited teams used multi-disciplinary models of practice, they were in the process of moving towards, or had an interest in, interprofessional models of care.

### 2.4. Data Collection

Qualitative data were collected via 11 focus group interviews (six with CEs and five with students) that took approximately one hour each. All focus groups were conducted via videoconferencing which allowed data collection across dispersed sites to continue during the COVID-19 pandemic. The focus groups were facilitated by one author (MF) trained in IPE and qualitative research. The CE and student focus groups were held separately to capture any unique perspectives held by each group and enable sharing of perspectives in a safe space. Appendix A and Appendix B present the focus group guides with questions used with CEs and students, respectively. Focus groups were audio-recorded and transcribed verbatim by an independent typist. A total of 579 min of recording was available for transcription.

### 2.5. Data Analysis

Consistent with the process for Qualitative Content Analysis [23], two researchers with experience in clinical education of students and qualitative research (AH and TB) engaged in multiple readings of the transcripts with a view to determining ‘what happened’ (i.e., the CE or student experience) and to appreciate participants’ views of those experiences. The unit of analysis was taken to be individual responses of participants. Open or deductive coding of half of the transcripts was undertaken by highlighting phrases within a unit of analysis that spoke to the research questions. A draft of the content categories was then applied to three transcripts with greater scrutiny via NVIVO (version 12) analysis software, allowing the draft content categories to be organised into a category and subcategory structure.

Following discussion and consensus building between the two involved researchers (AH and TB), further revisions to the content categories and subcategories was undertaken. Concise phrases, or codes, were added within subcategories to provide greater description (e.g., students were engaged in tutorials). As a further step to ensure rigour in the qualitative analysis process, two transcripts (one CE and one student) were coded by both involved researchers separately (AH and TB) and analysis/codes for each reference were compared side by side to determine agreement or disagreement. All disagreements were then discussed by the two involved researchers and a third researcher (PM) until consensus could be reached. The third researcher involved (PM) also reviewed the consistency of the dual-coded transcripts and contributed to the development of the final content categories and subcategories. The final content categories and subcategories were used to analyse all remaining transcripts via NVIVO (version 12) software. The same content categories were applied to both the student and CE transcripts; however, they were analysed separately in order to compare student and CE perceptions and experiences as part of the analysis.

### 2.6. Ethics

Ethics approval was obtained from the Darling Downs Health Human Research Ethics Committee (Reference number: LNR/19/QTDD/54776, Approval date: 4 September 2019), with subsequent site-specific approvals obtained from all involved health services. Written, informed consent was obtained from every focus group participant.

## 3. Findings

In total, there were 58 participants in the study: 24 CEs across four health services and 34 students from seven universities completing their placements. Only thirty-two responses were available for year level and previous interprofessional practice experience in a clinical setting (Table 1) due to a delay in obtaining site specific approval at one site impacting on data collection. Most of the students were on undergraduate placements (*n* = 28), with 88% (*n* = 29) of the students having been exposed to IPE concepts in their university courses. Seventy five percent (*n* = 18) of CEs and 56% (*n* = 19) of students participated in the focus groups. Further participant and placement characteristics are presented in Table 1. Sixty percent of the students had previous IPE experience in clinical settings. Content analysis of focus group data resulted in the development of the following categories: value of the RIPES placement model, unintended benefits to CEs, work units and rural areas, tension between uni-professional and IPE components, and sustainability considerations.

### 3.1. Value of the RIPES Placement Model

Both CE and student focus groups illuminated the value of the RIPES placement model to students, CEs, as well as the work units. Not only was the RIPES model described to be complementary to the uni-professional placement models of the involved professions, it was also found to be complementary to rural healthcare settings and teams.

#### 3.1.1. Complementary to the Uni-Professional Placement Model and Rural Sites

The RIPES placement model was considered by both CEs and students to complement the uni-professional student placement models from the included professions. One CE said:


*Even as I was going through the COMPASS [speech pathology clinical assessment tool] during the mid- and final evaluation, there was actually a lot that I could draw from the RIPES placement to help me actually mark the COMPASS, which was really good. And I think the students, at that point, too, started to realise that it [RIPES] wasn’t necessarily something that was additional on top of their university requirements, but it was actually something that married quite nicely together.*

*[CE 12]*


Both CEs and students noted that the RIPES model was complementary to rural healthcare settings, given the tightly knit nature of rural teams that was considered to work hand-in-hand with the nature of IPE. One student noted:


*Sixty-five [percent] being the rural set up and 35 [percent] of the placement being the RIPES set up. I don’t think the RIPES placement framework is the reason that they’re all in the same office. I think that’s just because that’s how the resources work here, but as [student name] said, the 35 [percent] comes from RIPES because you’re interacting with the other professions. Like, you’re forced to, it’s not just when you get the chance. You’re forced to go work shadow, talk to them, do all of that. It is that sort of innate setup of rural.*

*[Student 18]*


#### 3.1.2. Benefits of the Whole Model and the Individual Components

RIPES was considered beneficial overall to all stakeholders involved including students and CEs, as well as the work unit. Participants talked about the usefulness of RIPES in minimizing hierarchies in the workplace, and the value of individual components such as work shadowing and tutorials.

RIPES was seen as powerful in reducing inherent hierarchies experienced in healthcare teams. Both CEs and students felt that RIPES placed everyone in the team on a level playing field. One CE said:

When they [students] become clinicians, they will be far more rounded in knowing what the team does as an interprofessional entity rather than just what their role is within that team. And just having professional respect…I feel we work really well together because speech doesn’t think that they’re better than physio, physio doesn’t think they’re better than speech [CE 16]

Some students noted that they felt they could learn patient management skills earlier through the RIPES model, which was believed to influence better outcomes for patients. One student noted:


*From shadowing other professions, particularly OT, community health nurses etc., I found that when I go in to see a patient for the first time, my questions that I ask initially can screen a few of those things. If I know that they haven’t been referred to them, I can just get a quick idea “is this person going to need this person X, Y, Z reasons, and I can go there and go: Hey, guys, this person needs you guys”, and that their paths to going home or whatever starts a lot earlier than it would have, if I hadn’t had that shadowing.*

*[Student 1]*


The RIPES model was seen as a tool to enhance student readiness for work. Within the RIPES model, student IPE learning was noted to happen in a more structured way or faster than some CEs had experienced in prior roles or when compared to their own undergraduate clinical placements. CEs reported that team work or interprofessional behaviours (including relationship building skills) were able to be ‘fast tracked’ as an outcome of the RIPES model. Participants said:


*I think it [RIPES] has also helped with building relationships between the students themselves and also with us and the other team members….*

*[CE 17]*



*…the other key part is that you’re actually forming relationships with people from the other disciplines. You stop seeing them as an OT, you just see them as [name], who can do this.*

*[Student 18]*


In addition to finding the work shadowing helpful, students also commented on the usefulness of the weekly tutorials. One student while reflecting about work shadowing said:


*So, you’re a little bit more willing to go up and speak with them [staff], or ask them a question about a patient or if we’re trying to find out information about a patient…Whereas, prior to having that work shadowing I wouldn’t have, because I didn’t really know them. I think it makes you feel a little bit more included in the team.*

*[Student 7]*


Another student reflecting on the weekly IPE tutorial said:


*I quite like the tutorials, particularly the ones where everyone came and just spoke about their roles, and we had an open discussion. I found that one was very helpful. We both thought that we learnt a lot from that one.*

*[Student 19]*


### 3.2. Unintended and Flow-On Benefits to CEs, Patients and Rural Areas

An interesting finding was that CEs too, like students, reported various benefits of the RIPES model to their own learning, professional development and practice. CEs reported benefitting greatly themselves from better understanding of interprofessional competencies, which they were able to implement in their own practice. This was then seen to have a ripple effect that positively impacted patient care. Students, as a result of a positive placement experience, noted that RIPES confirmed their choice of a rural area for their future workplace location.

#### 3.2.1. Ripple Effects of the RIPES Model

Some CEs noted that the benefits they had reaped from facilitating the RIPES model with students was flowing on to their own practice and ultimately enhancing patient care. CEs said:


*I found the tutorials actually really beneficial for the clinicians as well. A lot of them were exploring individual roles but then the overlap and how we can work together. For my tutorial there was lots of feedback that I learned a lot from it. So, I don’t think it’s only for the benefit of the students. I think it’s helping us also.*

*[CE 2]*



*I think my interprofessional collaboration has been consolidated. I feel like with pediatrics (the students we had were paeds focused), the changes that I’ve seen in my practice is a stronger collaboration between OT and speech pathology when dealing with pediatric clients, more communication with other disciplines when it comes to clients in common. Those are the positives that I can see that have changed in my practice.*

*[CE 11]*


#### 3.2.2. Impact of the RIPES Model on Workforce Recruitment

RIPES model may be able to help address a workforce gap in rural areas by providing positive placement experiences for students that can influence their choice to return to work in those areas. Students who had a pre-placement intention to work rurally following graduation, confirmed that RIPES helped with cementing this intention. One student said:


*I probably found out second year onwards, I always wanted to work rural, but you do hear a lot of stories that a real generalist role can be a little overwhelming for a lot of people. I think since coming here and especially probably because we had RIPES, although you may not know anything about one particular case that you’ve never seen before, the teamwork I think is very proficient. I feel working rural would be a lot less lonely than what some people describe it as.*

*[Student 6]*


### 3.3. Tension between Uni-Professional and IPE Components

Some participants noted that there was a tension between the RIPES and the uni-professional placement requirements, mainly due to time commitments. Some students and CEs were concerned that RIPES participation detracted from their capacity to develop skills in their profession.


*From my perspective, sometimes it was a little tricky to get them the clinical opportunities or follow up like progress notes or something because practically Wednesday and Thursday afternoons were out for our students.*

*[CE 6]*


Some CEs noted the importance of student disposition and prior IPE experience as critical success factors for RIPES. In some instances, a CE’s interpretation of RIPES success was colored by their assessment of student capability. CEs noted that students who were motivated and engaged made observable gains in IPE competencies. Students who needed more development of their clinical skills or more support in a rural environment were considered unsuitable candidates for the RIPES model. This reflection was noted to come from CEs that were newer to the supervisory role, indicating that this may be reflective of where the CE was on their supervisory journey, as opposed to being solely about student capabilities. CEs said:


*Because we also have difficult students as well as the RIPES complexity. With the RIPES element of their placement the students really enjoyed it but it did add another layer of complexity to an already problematic placement. Certainly, their engagement with RIPES I don’t think was as good as it could be because they were difficult students.*

*[CE 9]*



*My student, she’s got six weeks, she has to be entry level at the end of this six weeks. … She’s a brilliant student. I’ve given her feedback and she learns and takes it on like that, but I know that’s not the case with a lot of students. So, I don’t think this would have been feasible if I had a student who was struggling, no way. In six weeks, you wouldn’t even try and put this extra stuff onto them.*

*[CE 15]*


### 3.4. Sustainability Considerations

Many CEs noted that they would continue implementing the RIPES model, although some CEs described conditions that would make implementation more practical or appropriate for students, or easier for CEs to share the work load. CEs said:


*So, that was one of the pressures, the fact that we’re trying to organize four students that are different disciplines, so trying to get our calendars to marry up sometimes is hard. Finding that time for them to do their project work was a bit difficult. Trying to organize other clinicians, too, because we had a number of guest speakers come in and trying to organize some time for shadowing was a little bit tough…Maybe if there were three clinicians involved and three different disciplines, I think that would be perfect. So, if one person was sick or had to go away, it wouldn’t just fall, the burden, on one person.*

*[CE 12]*



*I do think RIPES would be an excellent idea for that introductory, like a first placement that is. One of the first things that you need to learn is working as part of a team, communication and conflict resolution…they are skills that need to be learnt right up front at the start of pracs, not at the tail end.*

*[CE 14]*


## 4. Discussion

This study explored in-depth perspectives and experiences of CEs and students participating in a novel, interprofessional model of student placement, implemented in rural healthcare settings. Findings indicate that the RIPES model was valuable to all involved stakeholders including students and CEs, with notable flow-on effects to patient care and work units. RIPES was found to be complementary to uni-professional placement models related to the professions involved in the study (i.e., dietetics, occupational therapy, physiotherapy, and speech pathology), as well as complementary to rural healthcare settings. CEs reported several benefits to their own professional development and practice, an unintended outcome given the model was developed mainly to facilitate IPECP in students. It is reassuring that the model is also beneficial to post-qualification healthcare workers, reinforcing the important role of IPECP in healthcare settings [2,24]. The ripple effects that flowed-on to patient care and the work units are evidence of the benefits of the RIPES model that outlast the implementation period and permeate into other structures and processes related to patient care and service delivery at the implementation sites. While attempts to transition healthcare from a siloed to a collaborative approach are considered complex, challenging, and elusive [25], this study has provided evidence that investment in a collaborative approach is feasible and worthwhile for healthcare organisations.

The IPECP competencies from the Canadian Interprofessional Health Collaborative Framework (CIHC) [21] were used in the development of the RIPES model. Findings indicate that this framework is well-suited to rural healthcare settings and facilitates the development of IPECP competencies not only in students but also post-qualification healthcare workers. Rural areas in Australia rely on healthcare workers having a generalist skillset [18] to combat some of the well-known rural healthcare workforce challenges [26,27]. Given this context, rural healthcare workers need to possess teamwork, relationship building, collaborative practice and role clarification skills. Whilst the rural healthcare workforce can have a higher proportion of recent graduates, a model such as RIPES is well-placed to upskill the future healthcare workforce within the pre-qualification phase, so that they are better prepared for rural practice.

Whilst most CEs noted the value of integrating RIPES within regular placements and how complementary it was to the uni-professional placements, some CEs perceived the IPECP component as a luxury and as an add-on once the uni-professional placement competencies were met. Further awareness raising and education of the integral value of IPECP competencies may be needed so that more healthcare workers can value IPECP competencies as core to enhancing the work readiness of graduates. There was also a view that high performing students were more suited to RIPES, as opposed to those with perceived challenging behaviors. Healthcare worker interprofessional identity, although acknowledged to be important to progress IPECP, remains a black box [28]. Better knowledge of this concept, as it becomes available, may need to be incorporated into continuing professional development activities for healthcare workers on IPECP.

Sustainability is a well-known challenge in IPECP. To combat this, the study considered sustainability elements from the outset. All CEs involved in the RIPES study, their team leaders, and managers, were keen to continue implementing the RIPES model either in its entirety or with inclusion of some specific components. Logistical and scheduling challenges remain a major barrier to a model such as RIPES as aligning placement dates for students across professions involves several stakeholders from academia and healthcare settings, and multiple processes. Different universities also have different processes, even for the same profession. Unless academic and health service partners work more closely with each other, facilitating flexibility in placement dates, processes, and structures, sustaining a model such as RIPES will remain a mammoth challenge. While waiting for several planets to align, interested CEs can still facilitate components of RIPES with their students with practical tips now available to facilitate IPECP with a range of student placements (including a single student on placement) [29].

### Strengths and Limitations

This is the first known study to investigate in-depth experiences of CEs and students in implementing a novel interprofessional student placement model in a rural context across multiple sites, using the latest recommendations for best-practice IPECP research. A large-scale, phased, long-term research study of this nature, in rural areas, is usually not feasible given the resource constraints, high workforce turnover and pragmatic challenges. Therefore, this study has made a substantial contribution in a space and context that is hard to rigorously research and sustain. The academic-research partnership that drove this study was a critical success factor that also ensured that the study could continue despite the challenges posed by the COVID-19 pandemic. The study is not without its limitations. While every effort was made to recruit students and CEs from a wider range of professions (e.g., medicine and nursing), due to pragmatic, logistical and scheduling challenges this was impossible to achieve. Future research could investigate this, as well as the impact of the RIPES model on patient care, from a client perspective.

## 5. Conclusions

This study explored perspectives of CEs and students regarding the implementation of the RIPES student placement model in rural contexts. Students and CEs alike valued the learning which arose from participation in the model and the positive flow-on effects to both patient care and work units. Further research with a broader range of health professional students would be valuable to confirm the contribution of the RIPES model to enhancing students’ interprofessional competencies and prepare them for practice in rural contexts.

## Figures and Tables

**Table 1 ijerph-19-10734-t001:** Participant and placement characteristics.

	CEs (*n*)	Students (*n*)
Profession (*n* = 58):		
1. Dietetics	3	6
2. Occupational therapy	6	6
3. Physiotherapy	8	9
4. Speech pathology	7	13
Sites (*n* = 58):		
1. Site 1	2	2
2. Site 2	7	13
3. Site 3	2	6
4. Site 4	6	10
5. Site 5	5	4
Year level (*n* = 32):	n/a	
Undergraduate-Year 3:		4
-Year 4:		23
Masters-Year 2:		5
Previous Interprofessional Practice in a clinical setting (*n* = 32):	n/a	
No		13
Yes		19

CEs—Clinical Educators; n/a—not applicable.

## Data Availability

Data and materials are protected by ethics but may be made available in de-identified format upon contact made with the corresponding author.

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
