# Peer review of "A Novel Interprofessional Education and Supervision Student Placement Model: Student and Clinical Educator Perspectives and Experiences"

_ijerph, 2022, doi:10.3390/ijerph191710734_

Round 1
Reviewer 1 Report
Thank you for the opportunity to review this paper. Overall, I found this paper to be very interesting, and adding to the available literature on interdisciplinary clinical placements for nurses. I do have some minor comments/questions:
- The data presented is from a wider study where pre- and post-testing was conducted according to the manuscript. Was consideration given to presenting the study as a whole? I do feel that there is some background missing that could've been addressed by presenting the study as a whole.
- More background is required on the RIPES model. At present, the manuscript describes this as an interdisciplinary placement model but it would be of benefit to the reader to see detail beyond this description.
- Line 216 is presented as a quote but appears to be part of the previous paragraph.
- Section 3.2 would benefit from an exemplar quote to support the discussion.
- It would be good to see expansion on the difficult student/difficult placement data presented in section 3.3.
Overall, these are minor suggestions/queries, and I believe once addressed the paper would make a good addition to the journal.
Reviewer 2 Report
Comment: This is an interesting study on A novel interprofessional education and supervision student placement model and the authors have collected a unique focus group interview by using the cutting-edge methodology. The manuscript is generally well written and structured except the methods part. However, in my opinion, the article needs to revise of extensive English for better understanding and has some shortcomings in regard to some data analyses and text.
Below I have provided numerous remarks on the text and detailed suggestions in the whole document in line-by-line comments, as it is often the vague and improper argument of the findings. Furthermore, I made additional suggestions for the modification of some sentences.
Key critical points are
Abstract:
- Line 18: Please check the typo (Spelling) and correct it “owever”
- The authors are recommended to add the suggestions or recommendations for the implications of this study
Introduction:
- In the first paragraph: The authors suggested adding the statistics of professional ratio (All the medical professionals) in rural healthcare settings in Australia if available or adding the global statistics.
L 37: Recommended to cross-check the reference (WHO), it would be suggested to add the year of publication.
L. 61: please verify the RIPES model and provide a strong background of your study.
L 73: Clarify the (CIHC, xx), in which ‘xx’ stands for …..?
L. 77-79: Recommended to break the paragraph into two sentences for better understanding.
Methods:
The research methodology must be revised
* Authors are recommended to rearrange the orders as follows
L 81: 2.1 Research design à And 2.2. Setting
L 104-106: ‘All focus groups were conducted via videoconferencing’: please clarify whether focus group “education or interview”
L. 112-117: The recording and timings should be move into the data collection process.
Content analysis should be described in detail in the data analysis part.
Data analysis is supposed to begin from line no: 117: The unit of analysis……
Results:
L. 143: As per the research methodology heading must be named “ Results” or findings, please revise it.
Table 1 : N (%) of participants to be added. Add the total no. of participants in each focus group in the title “ Ces(n=?) %”
Also, the acronyms to be addressed under the table Eg: CEs= Clinical educators; n/a =?
